# In Vivo Anti-Inflammation Potential of *Aster koraiensis* Extract for Dry Eye Syndrome by the Protection of Ocular Surface

**DOI:** 10.3390/nu12113245

**Published:** 2020-10-23

**Authors:** Sung-Chul Hong, Jung-Heun Ha, Jennifer K. Lee, Sang Hoon Jung, Jin-Chul Kim

**Affiliations:** 1Natural Informatics Research Center, Korea Institute of Science and Technology (KIST), Gangneung 25451, Korea; schong@kist.re.kr; 2Research Center for Industrialization of Natural Neutralization, Dankook University, Cheonan 31116, Korea; ha@dankook.ac.kr; 3Department of Food Science and Nutrition, Dankook University, Cheonan 31116, Korea; 4Food Science & Human Nutrition Department, University of Florida, Gainesville, FL 32611, USA; leejennifer@ufl.edu; 5Natural Product Research Center, Korea Institute of Science and Technology (KIST), Gangneung 25451, Korea; shjung@kist.re.kr; 6Division of Bio-Medical Science and Technology, KIST School, Korea University of Science and Technology (UST), Daejeon 34113, Korea

**Keywords:** functional food, dry eye, *Aster koraiensis*, animal model and human retinal pigmented epithelial (ARPE-19) cells

## Abstract

Dry eye syndrome (DES) is a corneal disease often characterized by an irritating, itching feeling in the eyes and light sensitivity. Inflammation and endoplasmic reticulum (ER) stress may play a crucial role in the pathogenesis of DES, although the underlying mechanism remains elusive. *Aster koraiensis* has been used traditionally as an edible herb in Korea. It has been reported to have wound-healing and inhibitory effects against insulin resistance and inflammation. Here, we examined the inhibitory effects of inflammation and ER stress by *A. koraiensis* extract (AKE) in animal model and human retinal pigmented epithelial (ARPE-19) cells. Oral administration of AKE mitigated DE symptoms, including reduced corneal epithelial thickness, increased the gap between lacrimal gland tissues in experimental animals and decreased tear production. It also inhibited inflammatory responses in the corneal epithelium and lacrimal gland. Consequently, the activation of NF-κB was attenuated by the suppression of cyclooxygenase-1 (COX-1) and cyclooxygenase-2 (COX-2). Moreover, AKE treatment ameliorated TNF-α-inducible ocular inflammation and thapsigargin (Tg)-inducible ER stress in animal model and human retinal pigmented epithelial (ARPE-19) cells. These results prove that AKE prevents detrimental functional and histological remodeling on the ocular surface and in the lacrimal gland through inhibition of inflammation and ER stress, suggesting its potential as functional food material for improvement of DES.

## 1. Introduction

Dry eye (DE) has been identified as a common disease of the eye, which is induced by the failure of tear production and tear retention in the ocular surface [1]. Dry eye syndrome (DES), also known as keratoconjunctivitis sicca, is primarily caused by genetic factors, autoimmunity, and external environmental insults. A previous cohort study estimated that approximately 14.5% of the US population suffers from DE [2]. DE is mainly caused by the increased osmolality of the tear film and immune response on the ocular surfaces [3]. When tears lose their integrity or excessively evaporate, the tear film becomes thinner and unstable [4]. The weakened tear film results in damage with increased inflammatory responses on ocular surfaces. The tear film is the primary defensive line of the eye against foreign pathogens and abiotic factors, and it also maintains ocular homeostasis [5,6].

Furthermore, inflammation induced by an imbalance of ocular homeostasis triggers the recruitment of pro-inflammatory immune cells. Consequently, the lacrimal unit remodels its structures and the tear film properties [7,8,9]. Growing evidence suggests that DE-related ocular surface inflammation is mediated by immune cells [10]. Infiltrating inflammatory cells from peripheral circulatory blood were determined to secrete inflammatory cytokines such as interleukin (IL)-1, 6, 8, and 10 and tumor necrosis factor (TNF)-α on ocular surfaces. These cytokines degrade the corneal epithelial barrier and further induce apoptosis in the conjunctiva and lacrimal gland. Thus, an inflammatory response impedes the integrity of the ocular surfaces and accelerates the progression of DE [11,12,13].

The endoplasmic reticulum (ER) is a cellular organelle located in the cytoplasm. The ER has been known to metabolize proteins and lipids (biosynthesis, packaging, and secretion), and it stores calcium ions, which play a role as a cellular signal regulator. ER stress occurs by protein misfolding, including glycosylation or disulfide bond formation, protein overexpression, or mutations in protein s [14]. Severe intracellular Ca^2+^ dysregulations can promote cell death through apoptosis [15]. ER stress has received growing attention in many pathophysiological disorders, such as cardiovascular diseases, neurodegenerative diseases, inflammatory bowel disease, and rheumatoid arthritis [16]. Recently, it was suggested that ER stress is also related to DES [17].

Conventional therapeutic agents such as artificial tears, anti-inflammatory drugs, and corticosteroids have been determined to help DE patients in mitigating uncomfortable ocular symptoms and improving their clinical and pathological conditions for the short term [18,19,20]. However, this modality provides only temporary symptomatic relief, and up to two-thirds of DES sufferers complain of persisting symptoms despite such treatments [21]. Due to the limitations of medical treatment, a new therapeutic approach for DES is needed. An alternative method that can be presented is improvement through ingestion of functional food, which is a non-medical treatment method. Functional food derived from natural products was reported to have low side effects and to be safe for long-term intake [22,23,24]. In particular, it has preventive effects such as prevention of recurrence by improving the physical constitution during long-term ingestion [25]. In this study, *Aster koraiensis* extract (AKE) was selected as a candidate material of functional food to improve DES symptoms. *A. koraiensis*, also known as Korean starwort, is an herbaceous perennial plant of the Asteraceae family. Because *A. koraiensis* is widely distributed in most regions of Korea [26], it has been used as traditional herbal medicine and food item in the country [27,28]. *A. koraiensis* has been reported to inhibit diabetic inflammation [29,30] and enhance wound healing on the skin and during bronchitis [31]. Moreover, chlorogenic acid and 3,5-di-O-caffeoylquinic acid, both isolated from *A. koraiensis*, have been used to relieve symptoms of diabetic-related diseases [32,33].

In this study, we produced an extract to examine the improving effect of *A. koraiensis*, a medical edible herb crop in Korea, on DE. The functionality of *A. koraiensis* extract was evaluated in an animal model in which scopolamine-induced DE. To investigate the detailed mechanism of action, we confirmed the possibility of improving DE in *A. koraiensis* via animal tissues and human retinal pigmented epithelial (ARPE-19) cells. In this way, we try to prove the value that *A. koraiensis* can be used as a functional food material.

## 2. Materials and Methods

### 2.1. Plant Material and Preparation of A. koraiensis Extracts

*A. koraiensis* was collected from Jeongseon-gun, Gangwon-do, Republic of Korea, in September 2016. The plant’s aerial parts, including the flowers and leaves, were thoroughly washed with water to remove impurities and were further dried in the shade for 1 month. The medicinal plant mixture was extracted from the dried plants with ethanol (EtOH) by maceration at room temperature for 3 days. *A. koraiensis* ethanol extracts (AKE) were combined and concentrated by evaporation in a rotary evaporator at 60–70 °C. The EtOH extracts were then freeze-dried.

### 2.2. Antioxidative Activities of AKE

The antioxidative capacity of AKE related to either electron or radical scavenging was scrutinized with three different analytical methods and backgrounds; (1) A 1,1-diphenyl-2-picrylhydrazyl (DPPH), (2) the ferric reducing antioxidant power (FRAP), and the Trolox equivalent antioxidant capacity (TEAC). DPPH assay was performed to assess the stable DPPH radical generating capacity by AKE. FRAP assay was tested to understand reducing activity from ferric to ferrous iron by AKE. TEAC assay was intended to assess radical scavenging cation ABTS+ (2,2′-azinobis (3-ethylbenzothiazoline-6-sulfonic acid) by radical quenching or electron donation. Each evaluation was performed through the following experimental methods.

A 1,1-diphenyl-2-picrylhydrazyl (DPPH) assay was conducted as described by Serpen et al. [34] and Thaipong [35], with slight modifications. Briefly, 0.1 mL of 200 M DPPH reagent (Sigma-Aldrich Co., St. Louis, MO, USA) was added to 0.1 mL of each sample in 96-well plates. After incubation in the dark for 30 min, the absorbance was measured at 520 nm using a microplate reader (Spectramax M2E; Thermo Fisher Scientific, Waltham, MA, USA). Ascorbic acid (Sigma-Aldrich) as a DPPH-scavenging compound was used as a standard. Assay results were expressed in mg ascorbic acid/g.

The ferric reducing antioxidant power (FRAP) was assessed via the method developed by Benzie and Strain [36]; however, it had slight modifications. Briefly, acetic acid buffer (pH 3.6, 23 mM) was made by dissolving sodium acetate (Sigma-Aldrich) in acetic acid (Sigma-Aldrich). A 10 mM solution of 2,4,6-tripyridyl-s-triazine (TPTZ) was made by mixing 40 mM HCl (Sigma-Aldrich) and TPTZ (Sigma-Aldrich). A total of 25 μL of the AKE extract was then added to the FRAP reagent (acetic acid, 10 mM; TPTZ, 20 mM; FeCl_3_·6H_2_O = 10:1:1) and incubated in the dark at 37 °C for 15 min. The absorbance was measured at 593 nm using a microplate reader. FeSO_4_ (Sigma-Aldrich) was employed as a standard in this assay. Assay results were expressed in nM FeSO_4_/mL.

The Trolox equivalent antioxidant capacity (TEAC) was measured as described by Oki et al. [37] and Re et al. [38], with minor modifications. Briefly, 200 μL of 2,2′-azino-bis (3-ethylbenzothiazoline-6-sulfonic acid; ABTS) reagent (Sigma-Aldrich) was added to 10 μL of each sample in 96-well plates. After a 60-min incubation in the dark, the absorbance was measured at 405 nm using a microplate reader. The reaction rate was calibrated using Trolox equivalent (TE). Assay results were expressed in g TE/mL.

### 2.3. Total Polyphenol and Flavonoid Contents of AKE

Total polyphenol content was measured using the method described by Alves et al. [39], with minor modifications. Briefly, 10 μL of Folin–Denis reagent (Sigma-Aldrich) was added to 160 μL of each sample in 96-well plates. After 8 min, 30 μL of sodium carbonate (Showa Chemical Industry, Tokyo, Japan) was added to the mixture. After incubation in the dark for 2 h, the absorbance was measured at 765 nm using the Spectramax M2E microplate reader. Gallic acid (Sigma-Aldrich) was also used as the standard curve to extrapolate gallic acid equivalent (GAE). Results were expressed in mg GAE/mL.

Total flavonoid content was analyzed using slight modifications of the methods previously described by Pourmorad et al. [40] and Marinova et al. [41]. In brief, 400 μL of each sample was mixed with 1200 μL of EtOH (Junsei Chemical Co., Tokyo, Japan) and 240 μL of distilled water. Subsequently, 80 μL of 10% aluminum nitrate (Sigma-Aldrich) was added, followed by 80 μL of 1.0 M potassium acetate (Sigma-Aldrich). The mixture was allowed to stand in the dark for 40 min, and the absorbance was measured at 415 nm using the microplate reader. Quercetin (Sigma-Aldrich) was used to construct the standard curve against quercetin equivalents (QE), and the assay results were expressed in mg QE/mL.

### 2.4. Animal Experiment and Induction of DE Model

All procedures received approval from the Institutional Animal Care and Use Committee (IACUC) of Korea Institute of Science and Technology (KIST): KIST No. 2020-002, Gangneung Institute, and they were performed according to the Association for Research in Vision and Ophthalmology (ARVO) statement for the Use of Animals in Ophthalmic and Vision Research. Mice were accommodated in the animal room with air conditioning, temperatures of 22 ± 2 °C, humidity of 50 ± 10%, and 12 h light/12 h dark circadian cycles. Food and water were supplied ad libitum. Mice were acclimatized for 1 week and were later assigned into five groups composed of seven male 6-week-old BALB/c mice. Experimental DE in mice was achieved by twice-daily intraperitoneal (i.p.) injection of 200 μL (2.5 mg/mL) of phosphate buffered saline (PBS)-diluted scopolamine (Sigma-Aldrich). Groups of mice were orally administered with AKE once per day at concentrations of 0 (as vehicle control), 10, 50, or 100 mg/kg in 200 μL EtOH. In the control group, only PBS buffer was injected without scopolamine. In the case of AKE, 0 mg/kg in 200 μL EtOH was administered when orally administered as in the DE group. After 2 weeks, tear production was quantified by a standard Schirmer’s test strip placed in the lower one-third of the temporal eyelid before the eye was closed for 1 min. After the strip was removed, the length of the wet point was measured in millimeters in order to determine the Schirmer’s test value. For measurements of the tear breakup time (TBUT), 5 μL of sodium fluorescein was instilled into the eye and photographed. Corneal surface staining was performed to assess the extent of corneal surface changes. The corneal surface was observed and scored after the administration of one drop of 3% fluorescein (Sigma-Aldrich, St. Louis, MO, USA) into the inferior lateral conjunctival sac. The staining of the cornea was evaluated in a blinded manner. Mice are euthanized through cervical dislocation. When removing the cornea, click the mouse’s eyelid and use forceps to cut out the eyeball’s optic nerve that pops out and removes the cornea. After removing the crystalline lens inside the cornea, store in a freezer at −80 °C. After pulling the mouse’s lower jaw with forceps for removal of the lacrimal gland, cut off the epidermis of the pulled part so that the lower part of the mouse’s face can be seen. A lacrimal gland located near the lower jaw of the mouse can be secured. The secured lacrimal gland tissue is stored frozen at −80 °C.

### 2.5. Histology

The corneal epidermal tissue and lacrimal gland tissues were collected, fixed in 10% formaldehyde, and processed for paraffin embedding and sectioning. Sections were stained with hematoxylin and eosin (H & E) and examined a microscope (TE-2000U, Nikon, Tokyo, Japan) at X40. Central corneal epithelial thickness was evaluated in 5 sections for each cornea.

### 2.6. Western Blot Analysis

Total protein lysate was extracted from the corneal epidermal tissue and lacrimal glands of each mouse group and from APRE-19 cells after washing with ice-cold PBS (Invitrogen, Carlsbad, CA, USA) three times. Radioimmune precipitation assay (RIPA) buffer (Pierce, Rockford, IL, USA) was applied to the samples with protease inhibitors (Cell Signaling Technology, Danvers, MA, USA) and phosphatase inhibitor (PMSF, Thermo Fisher Scientific, Waltham, MA, USA) at 4 °C. After lysis, the supernatant was collected after centrifugation at 12,000× *g* for 15 min at 4 °C. Protein lysates were loaded and separated using SDS-PAGE gels (Bio-RAD, Hercules, CA, USA) and transferred to polyvinylidene difluoride (PVDF) membranes (Bio-RAD) using a wet-transfer method at 4 °C. Membranes were blocked with 5% non-fat milk in tris-buffered saline and Tween-20 (TBS-T) for 1 h to prevent nonspecific binding. Blots were incubated with primary and secondary antibodies at room temperature, developed with chemiluminescence reagent (Pierce), and detected and analyzed using LAS-4000 (General Electric Image Quant LAS 4000 Biomolecular Imager; GE Healthcare, Chicago, IL, USA) in grayscale. Experimental protein band intensity on blots was normalized to the intensity of glyceraldehyde 3-phosphate dehydrogenase (GAPDH), which did not vary significantly by treatment. Primary and secondary antibodies are summarized in Table 1.

### 2.7. Real-Time PCR (RT-qPCR)

Total RNA was isolated from corneal epidermal tissue, lacrimal glands, and ARPE-19 cells with the RNase mini kit (Qiagen, Valencia, CA, USA). Genomic DNA was removed by digestion with DNase I (Qiagen). Reverse transcription was performed using 1 µg of mRNA per sample with the RevertAid First Strand cDNA Synthesis Kit (Thermo Fisher Scientific, Waltham, MA, USA). Gene expression was assessed with a SYBR Green PCR mixture with gene-specific oligonucleotide primers using an AB 7500 real-time PCR machine (Thermo Fisher Scientific, Waltham, MA, USA). Primer sequences and parameters are described in Table 2. The expression of the target genes was normalized to the expression of β-Actin and glyceraldehyde 3-phosphate dehydrogenase (GAPDH), which was not significantly altered by treatments.

### 2.8. ARPE-19 Cell Culture

Human retinal epithelial ARPE-19 cells (American Type Culture Collection, ATCC) were cultured in DMEM/F-12 media (Gibco, Carlsbad, CA, USA) containing 10% FBS (HyClone Laboratories, Logan, UT, USA) and 1% penicillin/streptomycin (HyClone Laboratories). To induce an inflammatory and endoplasmic response in ARPE-19 cells, 2 × 10^5^ cells per well were cultured in 6 well plates. After culturing for 24 h, 200 μL media solution with 10 μg/mL of TNF-α or 5 μmol/L thapsigargin (Tg) was applied to the cells in the presence or absence of AKE (0, 0.1, 1, and 10 μg/mL) for 12 h. In the Control group case, the experiment was conducted with doubles without any inflammation-inducing factors or AKE.

### 2.9. Intracellular Calcium Release

Intracellular calcium [Ca^2+^]_i_ level was measured using a Fluo-4 NW calcium assay kit (F36206; Thermo Fisher Scientific, Waltham, MA, USA), according to the manufacturer’s protocol. Briefly, 1 × 10^5^ ARPE-19 cells per well were cultured in 96 well plates for 24 h, and ER stress was induced by the treatment with Tg and AKE (0, 0.1, 1, and 10 μg/mL) for 12 h. Cells were then incubated with a cell-permeable calcium indicator (Flow 4 A) for 1 hr before treatment of Tg (final concentration, 5 μmol/L). The [Ca^2+^]_i_ levels were accessed by measuring the fluorescent intensity using a microplate spectrophotometer (Bio-Tek Power Wave XS, Winooski, VT, USA). Image software (Bio-Tek software, Gen5) was applied for the subsequent quantitative analysis.

### 2.10. VEGF-α Secretion

1 × 10^5^ ARPE-19 cells per well were cultured in 96 well plates for 24 h. In order to measure the vascular endothelial growth factor (VEGF)-α secretion, Tg with AKE (0, 0.1, 1, or 10 μg/mL) were dissolved in the medium and treated with 100 μL each, incubated for 24 h, and then doubled and collected. VEGF-α protein secretion in the media was quantified using a commercial ELISA kit (BMS277-2; Invitrogen) as per the manufacturer’s instructions using a microplate spectrophotometer (Bio-Tek Power Wave XS, Winooski, VT, USA).

### 2.11. Statistical Analysis

Results are presented as means ± standard deviation (SD). Data were analyzed statistically using one-way ANOVA with Tukey’s post hoc analysis. For [Ca^2+^]_i_ determination, two-way ANOVA with Tukey’s post hoc test was used. Data were analyzed using a two-factor ANOVA test. If analytical results showed significant time X treatment interactions (*p* < 0.05), Tukey’s post hoc multiple comparisons test was applied. *p* < 0.05 was defined as statistically significant and was further indicated by a filled asterisk or number sign. All statistical analyses were performed using the GraphPad Prism 8 (San Diego, CA, USA).

## 3. Results

### 3.1. Antioxidative Effects and Polyphenol and Flavonoid Contents of AKE

Plants that are rich in secondary metabolites, including phenolics and flavonoids, have been identified to possess antioxidant properties afforded by their chemical structures and redox potentials [42]. As antioxidant activity is multifactorial and associated with several mechanisms [43], we performed three complementary tests to measure the antioxidant activity of AKE’s DPPH radical scavenging activity, ferric reducing antioxidant power (FRAP), and Trolox equivalent antioxidant capacity (TEAC). The AKE was prepared at different concentrations: 0.1, 0.5, 1, 5, and 10 mg/mL. As shown in Table 3, the DPPH radical scavenging activity of AKE at concentrations of 10, 5, and 1 mg/mL was determined to be significantly higher than at concentrations of 0.5 and 0.1 mg/mL (10 ≒ 5 ≒ 1 mg/mL > 0.5 ≒ 0.1 mg/mL). FRAP and TEAC were increased as the concentration of AKE increased in a dose-dependent manner. The total polyphenol content increased dose-dependently as the concentration of AKE increased (Table 4). The total flavonoid content increased as the concentration increased in the following order: 10 mg/mL > 5 mg/mL > 1 ≒ 0.5 ≒ 0.1 mg/mL). Thus, the amount of polyphenol and flavonoid is correlated with the antioxidant capacities. Moreover, the antioxidant activity is associated with a high concentration of extracts.

### 3.2. Effects of Aster koraiensis Ethanol Extracts on Eye Damage and Tear Production

In the scopolamine-induced mouse model of DE, scopolamine was observed to trigger the breakup of tear film, decrease tear production, irritate the lacrimal gland, and shrink the corneal epithelial cells [5]. To identify the effect of the AKE on DE, we administered AKE (0, 10, 50, or 100 mg/kg) orally once per day for 14 days to groups of mice with experimental DE. Eyes were stained with fluorescein in order to observe eye injury quantitatively. Under blue light, a green dot indicated the level of cornea damage (Figure 1A, black arrows). Quantitative analysis showed that AKE moderately ameliorated corneal damage in the DE mice in a dose-dependent manner (Figure 1B). The typical symptoms of DE include the quick dissipation of the tear film and a reduced amount of tear production [44]. To examine the mechanism by which AKE treatment inhibits corneal damage, tear breakup time (TBUT) and tear volume were observed. In DE mice, TBUT was reduced significantly, and AKE treatment at 100 mg/kg presented a reverse TBUT (Figure 1C). Additionally, the average volume of tear secretion was quantified using Schirmer’s test. AKE treatment moderately enhanced tear production in a dose-dependent manner (Figure 1D). Taken together, these data show that AKE inhibited scopolamine-inducible corneal damage with increases in tear production and tear film stability.

### 3.3. Histological Alterations of the Corneal Epithelial and Lacrimal Glands Following AKE Treatment in the DE Mouse Model

When DE symptoms develop, the corneal epithelial tissue thins to a delicate slim stratum [45]. In order to evaluate the effect of AKE on corneal epithelium from DE mice, we measured the thickness of corneal epithelial cells by histology. In Figure 2A, the dark pink area represents the corneal epithelial layer. This was significantly reduced in the DE group compared with the control (CON) group. AKE treatment inhibited the thinning of the corneal epithelial layer at doses of 50 and 100 mg/kg (Figure 2A,B).

Damage to the lacrimal glands is another general symptom of DE [46,47]. Lacrimal gland histopathology during AKE treatment of DE mice revealed a decrease in the number of infiltrating immune cells in the tissue, compared with the DE mice without AKE treatment (Figure 3C). Additionally, gaps in the glands were narrowed by the administration of AKE (Figure 3C). These results suggest that AKE is beneficial to corneal epithelial cells and lacrimal glands weakened during DE. Because these immune infiltrates are found to be common during DE, we next elucidated immune regulation by AKE treatment in experimental DE.

### 3.4. AKE Suppressed Immune Responses in the Corneal Epithelium and Lacrimal Glands of DE Model Mice

Inflammatory responses are closely related to the integrity of the ocular surfaces and structure of ocular units [7,13]. To investigate the role of an immune response during the pathology of DE, the relative mRNA expression levels of pro-inflammatory cytokines (interleukin 1 beta (IL-1β), Tumor necrosis factor alpha (TNF-α), Interferon gamma (IFN-γ), and Matrix metallopeptidase 9 (MMP-9)) were assessed in the corneal tissue and lacrimal gland from DE mice. All pro-inflammatory cytokines were found to be upregulated in the DE mice compared with the control mouse group. In the corneal tissues, AKE treatment inhibited the expression of each of these genes. In particular, IFN-γ and MMP-9 were remarkably attenuated, even at a low dose (10 mg/kg) of AKE (Figure 2C, Figure 3A). In lacrimal glands, AKE suppressed the expression of inflammatory cytokines IL-1β, TNF-α, IFN-γ, and TGF-β (Figure 3A). Especially, when 50 mg/kg of AKE was administered, cytokine expression did not decrease in proportion to the concentration in lacrimal gland tissues. However, AKE treatment experiment groups were found to reduce compared to the DE group that induced DE. Taken together, these data suggest that AKE treatment protected the eyes of the DE mice via effective anti-inflammatory activity in the corneal epithelium and lacrimal glands.

### 3.5. AKE Inhibited Inflammatory Response in Lacrimal Gland

To examine the molecular signaling pathway to be regulated by pro-inflammatory cytokines, the IκB/nuclear factor kappa B (NF-κB) axis, one of the main inflammations signaling pathways, was studied using western blot analysis of lacrimal gland tissue. Inactive NF-κB was observed to locate in the cytosol, bound with its inhibitor IκB. Phosphorylation of IκB by IκB kinases (IKKs) results in the degradation of IκB. Subsequently, NF-κB translocates to the nucleus, where it regulates gene expression concerning inflammation [48]. Optimal induction of the NF-κB target genes also requires the phosphorylation of NF-κB proteins, such as p65 [49]. AKE treatment has suppressed phosphorylation of IκB-α, implying a decrease in immune reaction. NF-κB has also showed a similar trend to that of IκB-α, but no significant difference between the control group and the DE group (Figure 3B). The target genes of the IκB/NF-κB axis increased in DE mice and were inhibited by AKE treatment (Figure 3B). Therefore, the immune response in lacrimal glands was markedly attenuated by AKE consumption.

### 3.6. AKE Inhibited Inflammatory Reaction and ER Stress on ARPE-19 Cells

Ocular inflammation has been known to occur not only on the corneal surface and lacrimal gland area in DE, but it also causes retinal damage. Two well-studied retinal damage-induced diseases are uveitis and diabetic retinopathy (DR) [50,51]. Thus, we observed the effect of AKE on a retinal damage model. The ARPE-19 cell line is a widely used cell line for eye study, while TNF-α is a pro-inflammatory cytokine that induces inflammation in ARPE-19 cells [52]. Treatment with AKE prevented TNF-α-inducible inflammation in ARPE-19 cells with attenuation of TNF-α, IL-1β, IL-6, IL-8, and MMP-9 mRNA expression (Figure 4A). In addition, post-translational modification by phosphorylation of mitogen-activated protein kinase (MAPK) was examined, which is a key pathway signal transduction in inflammation. Phosphorylation of p-38 and p-ERK proteins was markedly attenuated in AKE-treated ARPE-19 cells (Figure 4B, Appendix A). By contrast, retinopathy is closely related to increased ocular ER stress activated in the retina and retinal endothelial cells under diabetic and hypoxic conditions [53,54].

Given the potent anti-inflammatory properties of AKE, we asked whether AKE would be effective in preventing other ocular stresses, such as oxidative stress or ER stress. To address this question, we induced ER stress in ARPE-19 cells by treatment with Tg (5 μmol/L) in the presence or absence of AKE. VEGF-α has been identified as a proangiogenic factor involved in the pathophysiology of some ocular diseases with neovascularization [55]. The level of VEGF-α is a general marker of eye disease. At a 10 μg/mL dose, AKE significantly decreased Tg-inducible VEGF-α secretion into the media (Figure 4C). Loss of cellular homeostasis and disruption of Ca^2+^ signaling can lead to the activation of ER stress responses in both the reticular network and cytoplasmic compartments [56]. Tg treatment has remarkably increased [Ca^2+^]_i_ in ARPE-19 cells, implying that ER stress has also increased. AKE treatment decreased [Ca^2+^]_i_ release from ARPE-19 cells compared with the Tg group, suggesting that AKE may prevent Tg-induced ocular ER stress in ARPE-19 cells (Figure 4D). Taken together, AKE can attenuate inflammatory responses and Tg-inducible ER stress in human retinal epithelial cells.

## 4. Discussion

Dry eye syndrome (DES) has been defined as a chronic eye disease associated with aging, hormonal changes, inflammation, and autoimmune diseases [57]. The eyes of a patient with DE contain upregulation of inflammatory cytokines, such as IL-8, IL-6, IL-1β, TGF-β, and TNF-α, and infiltration of immune cells [58]. Therefore, the inhibition of an inflammatory reaction is one of the clinical strategies in treating DE [59]. Multiple inflammation-suppressive therapeutic agents, including cyclosporine A (immunosuppressive agent), corticosteroids (steroid), and tetracycline (anti-inflammatory drug), have been widely used in clinical practice to treat patients with DES. However, these agents only partly attenuate clinical symptoms; not all agents relieve the complete burden of DE symptoms. Thus, natural sources have been considered that produce fewer adverse side effects and promote easy usage of agents [60,61], and would be an alternative material for the treatment of DE symptoms.

*A. koraiensis* has been traditionally used as a treatment for diabetes and inflammatory-related diseases in Korea. A previous study has reported that *A. koraiensis* plays a vital role in retinal angiogenesis and oxidative stress-inducible retinopathy [33]. Inflammation-induced by ocular surface stress can lead to an immune response in order to generate additional damage and amplify inflammation [62]. In this study, we investigated whether AKE reduces ocular inflammation and ER stress. Our results demonstrated that oral administration of AKE to DE mice reversed the symptoms as it increased TBUT and improved tear production from lacrimal glands. Moreover, AKE treatment modestly augmented histological remodeling in the corneal epithelium and lacrimal glands from DE mice to a level similar to that of the control group. Atrophy of the lacrimal gland and loss of acinar cells leads to an osmotic imbalance in the tear film and propels the infiltration of inflammatory cells and expression of inflammatory cytokines, such as IL-1β, TNF-α, IFN-γ, and MMP-9, on the ocular surface and lacrimal glands [63,64,65]. The activation of NF-κB signaling was reported to stimulate inflammatory chemokines and cytokines, including IL-1β, IL-6, TNF-α, C-X-C motif chemokine 5 (CXCL5), C-X-C motif chemokine 8 (CXCL8), and monocyte chemoattractant protein 1 (MCP1) [66,67]. Additionally, IL-1β also activates COX-1 and COX-2, which are responsible for the synthesis of prostaglandins and contribute to the regulation of the inflammatory response [68].

Treatment with AKE (50 mg/kg) has restored tear production and acinar cells in the lacrimal gland of mice with experimental DE. Moreover, AKE significantly inhibited ocular inflammation in these mice through the downregulation of inflammatory mediators such as IL-1β, TNF-α, IFN-γ, and TGF-β at the transcriptional level in corneal tissue and lacrimal glands. It also suppressed the phosphorylation of i nhibitory κBa (IκBa) at a translational level. However, other inflammatory proteins, NF-κB, COX-1, and COX-2, did not alter dramatically, although AKE was effective in mitigating scopolamine-inducible DE in vivo. These findings suggest that AKE partially suppresses ocular inflammation and may affect other ocular stresses, such as oxidative and ER stress. Among the functional fractions of AKE, chlorogenic acids (CGAs) and 3,5-O-dicaffeoylquinic acids were identified to possess anti-inflammatory effects [69,70]. The anti-inflammatory response of AKE may rely on these functional chemicals. However, additional studies with specific functional components from the fractionated AKE should be performed to elucidate the active components of AKE.

Chronic ocular inflammation in DE also causes retinal damage. In a TNF-α-induced retinal inflammation (RI) model, AKE attenuated IL-1β, TNF-α, IL-6, IL-8, and MMP-9 in a dose-dependent manner. Along with NF-κB, the MAPK pathway plays a crucial role in inflammation, infiltration of innate immune cells, antigen presentation, and upregulation of inflammatory molecules [71,72]. Activation of the MAPK pathway was markedly attenuated in AKE-treated ARPE-19 cells. Together, these data suggest that AKE may reduce local inflammation in ocular tissues by suppressing the activation of inflammatory cytokines and the MAPK pathway. RI has been determined as a prominent signature in the pathogenesis of age-related macular degeneration, DR, and uveitis [73]. The anti-inflammatory effects of AKE need further investigation to reduce the burden of RI as well as DE.

ER stress is another primary intracellular event with chronic local inflammation [74]. In DR and retinal degeneration, increased ocular neovascularization and apoptosis were observed along with elevation of ER stress [75,76]. In the present study, AKE treatment was also determined to affect the inhibition of ER stress-related cellular pathologies. Additionally, we demonstrated that AKE has sufficient potency to inhibit the expression of VEGF-α and increase [Ca^2+^]_i_ levels in a model stress-triggered system. Previously, it was reported that VEGF-α and corneal lymphangiogenesis are increased in the DE mouse model [77]. Type 2 diabetes mellitus often triggers patients to have ophthalmic complications, including corneal abnormalities, glaucoma, iris neovascularization, cataracts, and neuropathies [78]. DR is the most common medical complication of these ophthalmic complications as it damages the blood vessels in the retina [79]. Indeed, DR is associated with increased intraocular levels of VEGF-α [78]. However, Shin et al. [80] reported that general nutrients and antioxidant bioactive materials such as polyphenolic substances in *A. koraiensis* possessed radical scavenging activity and reducing power. The current report promotes the study of the antioxidant ability by AKE as one of the underlying mechanisms to inhibit DE. As summarized in Table 3 and Table 4, a high concentration of AKE was associated with greater antioxidative activities. A study investigating the protective effects of muscadine grape polyphenols (MGPs) demonstrated that the major polyphenols (quercetin, ellagic acid, myricetin, and kaempferol) in MGPs effectively attenuated ocular inflammation and ER stress [81]. In the present study, we demonstrated that AKE mitigated symptoms of DE, including corneal epithelium thinning, lacrimal gland tissue gap formation, and tear production decline. The underlying mechanism is that AKE treatment inhibited inflammatory responses to attenuate the activation of IκB. Moreover, AKE treatment ameliorated retinal inflammation and ER stress.

## 5. Conclusions

In conclusion, the oral administration of AKE could be an effective treatment for DES or inflammation caused by ocular disorders. Especially, substances derived from natural products are in the limelight as raw materials for functional foods in that they can be easily processed as raw materials for food, and their safety is generally guaranteed. Therefore, if it is used as a functional material that anyone can easily accept and as a material for health foods for improving eye health, it can be judged that it is beneficial for reducing inflammation and alleviating DE. However, before it can be used as an actual food ingredient, evaluating its health effects in a human test, such as repetition and genetic toxicity, is required. If we do more of these studies, we can expect that they can be used as a convenient natural resource to improve DES.

## Figures and Tables

**Figure 1 nutrients-12-03245-f001:**
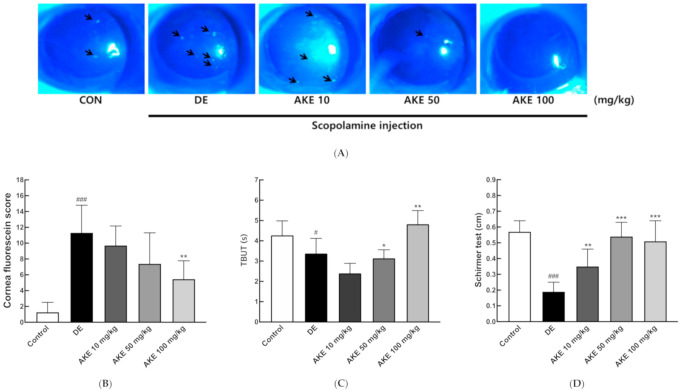
Effects of *A. koraiensis* ethanol extracts (AKE) on eye damage and tear production. Dry eye (DE) was reportedly induced by scopolamine injection, and AKE was administered at 0 (control; CON which was not induced by scopolamine injection, DE), 10 (AKE 10), 50 (AKE 50), or 100 mg/kg (AKE 100). (**A**) Representative images of corneal fluorescein staining. Black arrow indicates ocular damage spots of corneal fluorescein staining. (**B**) Quantitative analysis of images in A. (**C**) Tear breakup time (TBUT) was measured in seconds and analyzed. (**D**) Tear volume was measured using Schirmer’s test. In the graph, each bar represents mean ± SD of n = 7 mice per group. ^#^
*p* < 0.05, ^###^
*p* < 0.001 versus CON; * *p* < 0.05, ** *p* < 0.01, *** *p* < 0.001 versus DE. Data were analyzed statistically using one-way ANOVA followed by Tukey’s post hoc test.

**Figure 2 nutrients-12-03245-f002:**
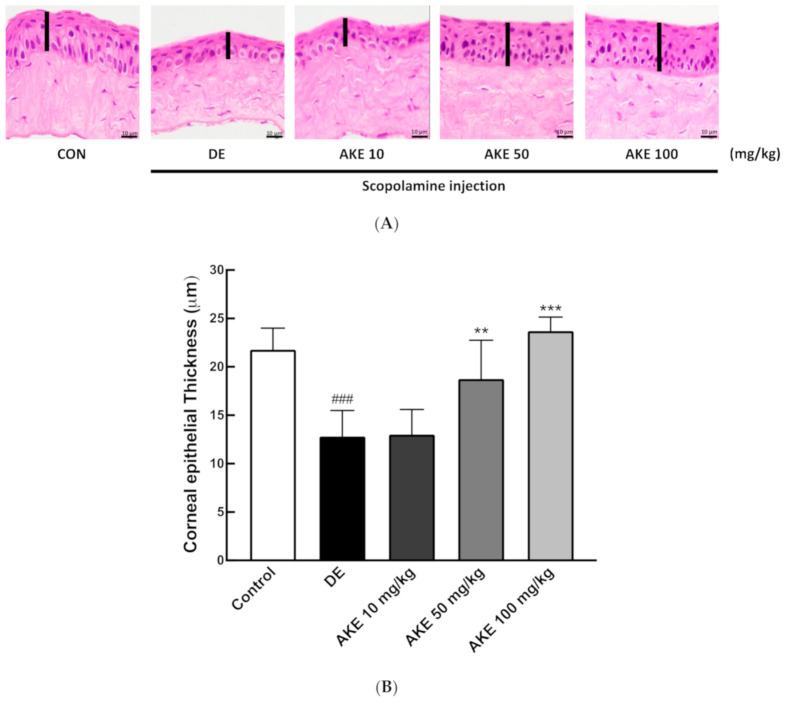
Effects of *A. koraiensis* ethanol extracts (AKE) on the corneal epithelial thickness and inflammatory cytokine expression. Dry eye (DE) was reportedly induced by scopolamine injection, and AKE was administered at 0 (control; CON which was not induced by scopolamine injection, DE), 10 (AKE 10), 50 (AKE 50), or 100 mg/kg (AKE 100). (**A**) Representative H and E staining images of histological sections of corneas. (**B**) Quantitative analysis of A. (**C**) Expression of inflammatory cytokines was accessed by real-time PCR (RT-qPCR) for each target gene (IL-1β, TNF-α, IFN-γ, and MMP-9). β-actin was used as an internal control. Each bar represents the mean ± SD of *n* = 7 mice per group. ^##^
*p* < 0.01, ^###^
*p* < 0.001 versus CON; ** *p* < 0.01, *** *p* < 0.001 versus DE. Data were analyzed statistically using one-way ANOVA followed by Tukey’s post hoc test. IL-1β, interleukin 1 beta; TNF-α, Tumor necrosis factor alpha; IFN-γ, Interferon gamma; MMP-9, Matrix metallopeptidase 9.

**Figure 3 nutrients-12-03245-f003:**
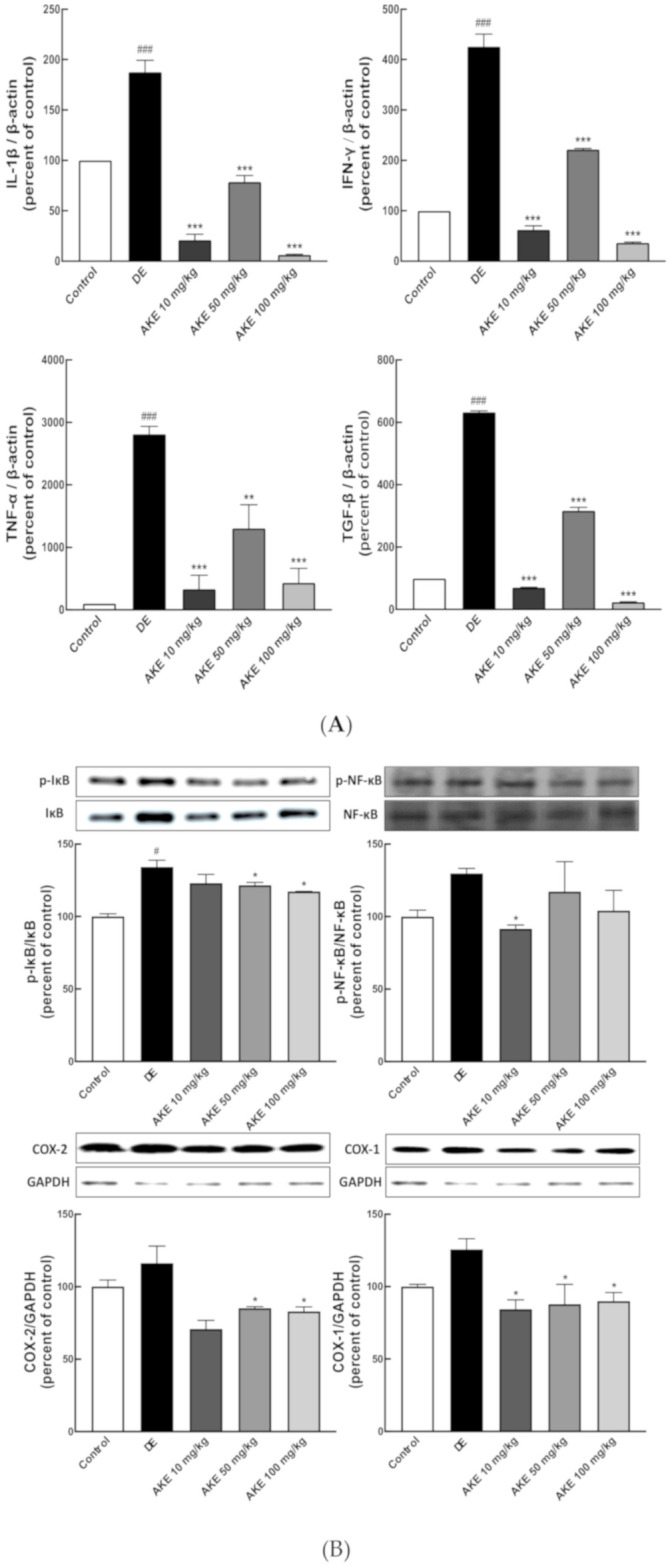
Effects of *A. koraiensis* ethanol extracts (AKE) on the inflammatory response and gap between tissues in the lacrimal gland. Dry eye (DE) was reportedly induced by scopolamine injection, and AKE was administered at 0 (control; CON which was not induced by scopolamine injection, DE), 10 (AKE 10), 50 (AKE 50), or 100 mg/kg (AKE 100). (**A**) Expression of inflammatory cytokines was accessed using RT-qPCR for each target gene (IL-1β, TNF-α, IFN-γ, and Transforming growth factor beta (TGF-β)). β-Actin was used as an internal control. (**B**) Inflammatory proteins were analyzed by western blot (WB). (Upper panels) Representative figures of WB for phosphorylated inhibitor of nuclear factor kappa B (IkB) and total IkB, phosphorylated nuclear factor kappa B (NF-κB), and total NF-kB were targeted. Quantitative analysis revealed the ratio of phosphorylated protein/total protein. (Lower panels) The representative figures and their analyses of WB for cyclooxygenase (COX)-1 and COX-2. GAPDH was used for the loading control. (**C**) Representative H & E-stained histological sections of lacrimal glands. The white area in the image shows the gaps between tissues in the lacrimal gland. Each bar represents the mean ± SD of n = 7 mice per group. ^#^
*p* < 0.05, ^###^
*p* < 0.001 versus CON; * *p* < 0.05, *** *p* < 0.001 versus DE. Data were analyzed statistically using one-way ANOVA followed by Tukey’s post hoc test.

**Figure 4 nutrients-12-03245-f004:**
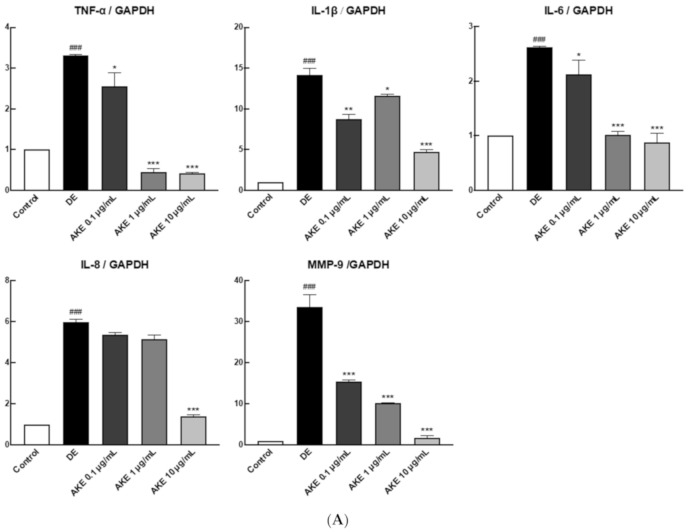
Effects of *A. koraiensis* ethanol extracts (AKE) on ER stress and the inflammatory response in animal model and human retinal pigmented epithelial (ARPE-19) cells. TNF-α-inducible pro-inflammatory gene expression in AKE-treated ARPE-19 cells. To induce an inflammatory reaction, phosphate buffered saline (PBS) or 10 μg/mL of TNF- α was treated, and AKE was treated in a dose-dependent manner (0, 0.1, 1, and 10 mg/mL). (**A**) Inflammatory cytokines were measured by qRT-PCR in order to validate the expression of each target gene (TNF-a, IL-1b, IL-6, IL-8, and MMP-9). glyceraldehyde 3-phosphate dehydrogenase (GAPDH) was used as an internal control. (**B**) The intracellular inflammatory signaling (MAPK) was analyzed using western blot (WB) with specific antibodies such as phospho-c-Jun N-terminal kinase (p-JNK), JNK, p-P38, P38, phospho-extracellular signal regulated kinase (p-ERK), and ERK. GAPDH was used for loading control. (C and D) endoplasmic reticulum (ER) stress was induced in ARPE-19 cells by thapsigargin (Tg), and AKE was treated with 0 (control; CON), 0.1 μg/mL (AKE 0.1), 1 μg/mL treatment (AKE 1), and 10 μg/mL treatment (AKE 10). (**C**) Tg-induced VEGF-α expression was measured using ELISA. (**D**) Tg-induced calcium ion efflux measurement using fluorescence. Data from three independent experiments have been presented as bar graphs showing mean ± SD. ^###^
*p* < 0.001 versus CON; * *p* < 0.05, ** *p* < 0.01, *** *p* < 0.001 versus DE. Data were analyzed statistically using one-way ANOVA, followed by Tukey’s test. For [Ca^2+^]_i_ determination, two-way ANOVA with repeated measures was used, followed by Tukey’s hoc test. A, C, D) Negative control to DE.

**Table 1 nutrients-12-03245-t001:** Antibodies for western blot analysis.

	Antibody	Dilution Factor	Corporation
Primary antibody	phospho-p65	1:1000	Cell signaling
	p65	1:1000	Cell signaling
	phospho-ERK	1:1000	Cell signaling
	ERK	1:1000	Santa Cruz
	phospho-JNK	1:1000	Cell signaling
	JNK	1:1000	Santa Cruz
	phospho-p38	1:1000	Cell signaling
	p38	1:1000	Santa Cruz
	phospho-AMPK	1:1000	Cell signaling
	AMPK	1:1000	Santa Cruz
	phospho-IκB	1:1000	Cell signaling
	IκB	1:1000	Santa Cruz
	COX-1	1:1000	Cell signaling
	COX-2	1:1000	Cell signaling
	GAPDH	1:2000	Cell signaling
Secondary antibody	Goat anti-mouse-HRP	1:2000	Santa Cruz
	Goat anti-rabbit-HRP	1:5000	Thermo scientific

ERK, extracellular signal regulated kinase; JNK, c-Jun N-terminal kinase; AMPK, 5’ adenosine monophosphate-activated protein kinase; IκB, inhibitor of nuclear factor kappa B; COX-1, cyclooxygenase-1; COX-2, cyclooxygenase-2; GAPDH, glyceraldehyde 3-phosphate dehydrogenase; HRP, horseradish peroxidase.

**Table 2 nutrients-12-03245-t002:** RT-qPCR primer sequences (5ʹ to 3ʹ).

Transcript	Forward Primer	Reverse Primer	Annealing Temp. (°C)
IL-1β	TCATTGTGGCTGTGGAGAAG	GGTGTGCCGTCTTTCATTAC	53.8
TNF-α	CTCAGATCATCTTCTCAA	CAGAGCAATGACTCCAAA	55.1
TGF-β	GAAAGCCCTGTATTCCGTCTCCTT	CAACAATTCCTGGCGTTACCTTGG	53.8
IFN-γ	AGCGGCTGACTGAACTCAGATTGTA	GTCACAGTTTTCAGCTGTATAGGG	61.2
IL-6	TTCCCTACTTCACAAGTC	GGTTTGCCGAGTAGATCT	52.9
IL-23	CAAGCAGAACTGGCTGTTGTC	GCACCAGCGGGACATATGAA	60.2
MMP-9	CACAACCGACGACGACGAGTTGTG	CTGTGGTGAGGCCGAATAG	65.0
β-Actin	TTGTTACCAACTGGGACGACATGG	GATCTTGATCTTCATGGTGCTAGG	59.2
GAPDH	ATGGTGAAGGTCGGTGTG	ACCAGTGGATGCAGGGAT-	58.0

IL-1β, interleukin 1 beta; TNF-α, Tumor necrosis factor alpha; TGF-β, Transforming growth factor beta; IFN-γ, Interferon gamma; IL-6, interleukin 6; IL-23, interleukin 23; MMP-9, Matrix metallopeptidase 9.

**Table 3 nutrients-12-03245-t003:** DPPH radical scavenging activity, FRAP, and TEAC of *A. koraiensis* ethanol extracts (AKE).

	0.1 mg/mL	0.5 mg/mL	1 mg/mL	5 mg/mL	10 mg/mL
DPPH radical scavenging activity (mg ascorbic acid/g)	30.8 ± 2.55 ^b^	80.0 ± 3.05 ^b^	112.7 ± 3.02 ^a^	117.8 ± 0.58 ^a^	118.4 ± 0.55 ^a^
FRAP (nM FeSO_4_/mL)	0.2 ± 0.01 ^e^	1.0 ± 0.02 ^d^	2.0 ± 0.10 ^c^	7.8 ± 0.41 ^b^	9.7 ± 0.07 ^a^
TEAC (g TE/mL)	0.2 ± 0.01 ^e^	0.4 ± 0.02 ^d^	0.9 ± 0.10 ^c^	2.7 ± 0.77 ^b^	2.9 ± 0.01 ^a^

AKE, *A. koraiensis* ethanol extracts; DPPH, 2,2-diphenyl-1-picrylhydrazyl; FRAP, ferric reducing antioxidant power; TEAC, Trolox equivalent antioxidant capacity; TE, Trolox equivalent. Values are means ± SD, n = 4. Data were analyzed by one-way ANOVA analysis followed by Tukey’s posthoc test. Means labeled without a common letter differ, *p* < 0.001 (DPPH radical scavenging activity) and *p* < 0.01 (FRAP and TEAC).

**Table 4 nutrients-12-03245-t004:** Total polyphenol and flavonoid contents of *A. koraiensis* ethanol extracts (AKE).

	0.1 mg/mL	0.5 mg/mL	1 mg/mL	5 mg/mL	10 mg/mL
Total polyphenol content (mg GAE/mL)	0.2 ± 0.00 ^e^	0.6 ± 0.01 ^d^	0.9 ± 0.01 ^c^	1.2 ± 0.01 ^b^	2.6 ± 0.03 ^a^
Total flavonoids content (mg QE/mL)	2.2 ± 0.58 ^c^	5.8 ± 1.37 ^c^	15.5 ± 0.61 ^c^	32.7 ± 1.51 ^b^	336.9 ± 11.06 ^a^

AKE, *A. koraiensis* ethanol extracts; GAE, 2 gallic acid equivalent; QE, quercetin equivalents; Values are means ± SD, n = 4. Data were analyzed by one-way ANOVA analysis followed by Tukey’s post hoc test. Means labeled without a common letter differ, *p* < 0.01 (polyphenol and flavonoid).

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
