# Peer review of "In Vivo Anti-Inflammation Potential of Aster koraiensis Extract for Dry Eye Syndrome by the Protection of Ocular Surface"

_nutrients, 2020, doi:10.3390/nu12113245_

Round 1
Reviewer 1 Report
Thank you for this interesting study based upon the first treatment of experimental DES with Aster koraiensis extract. My comments are as follows:
Thank you for this interesting study based upon the first treatment of experimental DES with Aster koraiensis extract.
My comments are as follows:
Introduction:
-line 40: Consider remove “also known as keratoconjunctivitis sicca”: KCS is one of the expressions of DES, the term is not a synonym.
Discussion:
-line 371-372: consider adding the effects on DES of systemically administered essential fatty acids: Clinical trials have demonstrated that in humans supplementation with either ω-3 or ω-6 essential fatty acids, or both, improved subjective symptoms, eyelid margin hyperemia, tear break-up time and tear secretion. [Yagci A, Gurdal C. The role and treatment of inflammation in dry eye disease. International Ophthalmology 2014; 34: 1291-1301. Alves M, Fonseca EC, Alves MF, et al. Dry eye disease treatment: a systematic review of published trials and a critical appraisal of therapeutic strategies.], and the effects on KCS of a periocular fatty acids application: Preliminary results of a pilot study indicated that the application of periocular FAG and topical 0.15% hyaluronate eye drops improved clinical signs of KCS in dogs. [The Effect of Periocular Fatty Acids and 0.15% Hyaluronate Eye Drops Application on Keratoconjunctivitis Sicca in Dogs: An Exploratory Study Amalfitano C et al. Topics in Companion Animal Medicine 35 (2019) 18-25].
Author Response
- Line 40: Consider remove “also known as keratoconjunctivitis sicca”: KCS is one of the expressions of DES, the term is not a synonym.
- We removed “also known as keratoconjunctivitis sicca” per reviewer’s comment.
- line 371-372: consider adding the effects on DES of systemically administered essential fatty acids: Clinical trials have demonstrated that in humans supplementation with either ω-3 or ω-6 essential fatty acids, or both, improved subjective symptoms, eyelid margin hyperemia, tear break-up time and tear secretion. [Yagci A, Gurdal C. The role and treatment of inflammation in dry eye disease. International Ophthalmology 2014; 34: 1291-1301. Alves M, Fonseca EC, Alves MF, et al. Dry eye disease treatment: a systematic review of published trials and a critical appraisal of therapeutic strategies.], and the effects on KCS of a periocular fatty acids application: Preliminary results of a pilot study indicated that the application of periocular FAG and topical 0.15% hyaluronate eye drops improved clinical signs of KCS in dogs. [The Effect of Periocular Fatty Acids and 0.15% Hyaluronate Eye Drops Application on Keratoconjunctivitis Sicca in Dogs: An Exploratory Study Amalfitano C et al. Topics in Companion Animal Medicine 35 (2019) 18-25].
- We appreciate your interesting comment on our paper. It is great point on the effective treatment of DES by addition of PUFAs. However, in this study we would like to focus and emphasize the effects of AKE. Unless AKE is fat soluble components, therefore, addition of PUFAs may not enhance absorption, distribution and metabolism of AKE. However, in further study, we may deeply consider the co-administration of AKE with PUFAs for synergistic effects in the prevention of DES.
Reviewer 2 Report
Manuscript nutrients-958875 explores the potential use of Aster koraiensis as material in the development of a functional food with protective properties on the ocular surface through inhibition of inflammation and endoplasmic reticulum stress. I suggedt aceept it after minor revisions.
L24: Please define what is ER
L72: Please added more references in order to support the affirmation. Here there are some examples:
- Peptides. 122, 170170. https://doi.org/10.1016/j.peptides.2019.170170
- Annual Review of Food Science and Technology. 11: 93-118. https://doi.org/10.1146/annurev-food-032519-051708
L92: Please added a short sentence about the importance of use different antioxidant methods.
L136: Please added more information about how the animals were assigned to different treatments? I mean, those animals were randomized? What methodology of randomization was follow?
L 224, L230. Table 3 and 4. Please added the information about the numbers (0.1, 0.5, 1, 5, 10). I mean, I know that are the different concentration of AKE, but is in order to facilitate the comprehension and clarity for the reader
Author Response
Manuscript nutrients-958875 explores the potential use of Aster koraiensis as material in the development of a functional food with protective properties on the ocular surface through inhibition of inflammation and endoplasmic reticulum stress. I suggest accept it after minor revisions.
- L24: Please define what is ER
- We apologize for not providing enough information. ER is specified as endoplasmic recticulum. Furthermore, we created the definition in line 24.
- L72: Please added more references in order to support the affirmation. Here there are some examples:
- 122, 170170. https://doi.org/10.1016/j.peptides.2019.170170
- Annual Review of Food Science and Technology. 11: 93-118. https://doi.org/10.1146/annurev-food-032519-051708
- We are very grateful for the reviewer's feedback on the additional references. Two references have been added to reflect the views of the review (line 72 in the revised manuscript).
- L92: Please added a short sentence about the importance of use different antioxidant methods.
- We are very grateful for the advice that can improve the quality of this paper. As recommended by the review, we have made additional statements about the antioxidant experiments' purpose and background. The antioxidative capacity of AKE related to either electron or radical scavenging was scrutinized with three different analytical methods and backgrounds; 1) A 1,1-diphenyl-2-picrylhydrazyl (DPPH), 2) the ferric reducing antioxidant power (FRAP), and the Trolox equivalent antioxidant capacity (TEAC). DPPH assay was performed to assess the stable DPPH radical generating capacity by AKE. FRAP assay was tested to understand reducing activity from ferric to ferrous iron by AKE. TEAC assay was intended to assess radical scavenging cation ABTS+ (2,2′-azinobis(3-ethylbenzothiazoline-6-sulfonic acid) by radical quenching or electron donation. Each evaluation was performed through the following experimental methods.
- L136: Please added more information about how the animals were assigned to different treatments? I mean, those animals were randomized? What methodology of randomization was follow?
- We apologize for the insufficient description of the methodology for animal selection process. The animal selection process did not select based on a random methodology. In order to maintain the same animal condition as much as possible, animal experiments were conducted under the following conditions. We conducted experiments under the same conditions for all factors that can control the company that supplies the animals, the time of birth of the animals, and the identity of body weight and gender.
- L 224, L230. Table 3 and 4. Please added the information about the numbers (0.1, 0.5, 1, 5, 10). I mean, I know that are the different concentration of AKE, but is in order to facilitate the comprehension and clarity for the reader
- We apologize for the inadvertent mistake and truly appreciate your meticulous review. We have completed the modifications to both of the tables pointed out for the readability of the reader (line 248,254 in the revised manuscript).
Reviewer 3 Report
Manuscript ID: nutrients-958875
Type of manuscript: Article
Title: In vivo anti-inflammation potential of Aster koraiensis extract for dry eye syndrome by the protection of ocular surface
Authors: Sung-Chul Hong, Jung-Heun Ha, Inhae Kang, Sang Hoon Jung, Jin-Chul Kim *
Thank you for giving me an opportunity to review this manuscript by Hong et al. In this manuscript the author has shown anti-inflammatory properties of Aster koraiensis against dry eye syndrome in vivo and in vitro model and thus can be used as an effective treatment for dry eye syndrome. I enjoyed reading the manuscript. However, I have some suggestions and questions that need to be addressed and should be included to increase the strength of the paper.
- In the introduction section, the hypothesis and aim of the study should be clearly stated.
- The materials and methods section should be written in more detail for example how much the dried-plant was used for extraction, how much volume of ethanol was used for maceration, what was the volume of koraiensis ethanol extracts (AKE) was used at the beginning, and how much was at the end after concentrating by evaporation.
- Within the animal experiment and induction of DE model section under Materials and methods, please mention clearly about all the five groups as it doesn’t say anything about the control group shown in all the figures and tables. Please mention clearly what is this control group as the author only mentions that Groups of mice were orally administered with AKE once per day at concentrations of 0 (as vehicle control), 10, 50, or 100 mg/kg. I assume that this vehicle group is DE. Please correct.
- In the material and method section, the author did not mention how the corneal epidermal tissue and lacrimal glands were collected from mice. Please mention it in detail.
- Please mention how many in vivo and in vitro experiments were performed and whether the in vitro experiments have been done in duplicate or triplicate in the method section and legend?
- In the histology section, it doesn’t mention anything about how the epithelial layer thickness has been measures and quantified (Figure 2A and B), how many areas were selected, single or both eyes were used while quantification, etc. Also, it does not mention how the corneal fluorescence staining was quantified to show corneal damage (shown in Figure 1A and B). Please mention all these things in detail.
- For the western blot experiment, how many ARPE-19 cells how were used for treatment (TNF/Tg and AKE) in what volume? How long they have been cultured before experiment/treatment? What was used in the control group (mention about vehicle)? At the end of the treatment how the cells were collected for protein extraction? How much protein was used for western blot? Please mention all these in detail.
- How many cells were used for Intracellular calcium release, and VEGF-α secretion experiments and for the QRT-PCR experiment? How long the cells have been before (TNF/Tg and AKE) treatment in what volume? Please mention.
- At line 163 it says α-tubulin or GAPDH, while in the whole manuscript only GAPDH has been shown for western blot. Please correct.
- In Statistical analysis, please mention in detail which data has been used to analyze by which method.
- In Figure 1, please explain the control group (if it is without DE). In legend, it says that AKE was administered at 0 (control; CON), but it seems the DE group has been administered AKE (0, 10, 50, and 100). Please mention clearly about the control group if the mice in the control group were injected PBS by intraperitoneal injection and then given vehicle orally (mention if ethanol). Also please mention the volume used for oral administration.
- In Figure 1B, please mention the parameter measured/quantified on the y-axis along with the unit of measurement. In Figure 1C, please mention if TBUT is measured in second in legend.
- In Figure 2A, it would be nice to show a line to show the epithelial layer. In Figure 2B, please write corneal epithelial thickness (um) on the y-axis.
- At lines 263-266 it says “Lacrimal gland histopathology during AKE treatment of DE mice revealed a decrease in the number of infiltrating immune cells in the tissue, compared with the DE mice without AKE treatment (Figure 2C). Additionally, gaps in the glands were narrowed by the administration of AKE (Figure 2C)”. It does not match with the data shown in Figure 2C (which seems to be mRNA levels of various cytokines in corneal extract). Please correct.
- Figure 3A and B does not match with the text mentioned under AKE suppressed immune responses in the corneal epithelium and lacrimal glands of DE model mice. Also, mention Figure 3C in the result section. Please correct.
- For the QRT-PCR quantification data (Figure 2C, Figure 3A, and Figure 4A) please write if it is a relative expression of the mRNA gene for specific cytokines or mRNA levels as fold change on the y-axis to make it more clear. Please correct in the graphs and legends.
- Please mention for all the western blot quantification results on the y-axis if it is a ratio of the expression level of protein of interest to the housekeeping gene on the graph and in legend.
- It would be nice to show the quantification data for the ratio of p-JNK/JNK, p-P38/P38, and p-ERK/ERK for Figure 4B in addition to the western blot image.
- Based on the materials and methods section it seems that Intracellular calcium level was performed in vitro, while in Figure 4D it mentions n = 7 mice per group. Please correct.
- At line 310, it is not clear what is “3.5 AKE inhibited inflammatory response in lacrimal gland”. Please correct.
- At line 38 DE should be written as Dry eye (DE), please correct.
- At line 84 make A. koraiensis italics.
- Please mention the legend below Table 4 (as shown in Table 3).
- At lines 175-176 it says that “The expression of the target genes was normalized to the expression of GAPDH”, while at lines 278 and 301 it says β-Actin was used as an internal control for QRT-PCR, and at line 355 it says Gapdh was used as an internal control. Please correct and modify Table 2 accordingly.
- In discussion it would be better not to write Figure as () at the end of the sentence at line 383, 385, 393, 394, 408, 411, and 421.
- The authors should also discuss the limitations of the present study.
Author Response
Thank you for giving me an opportunity to review this manuscript by Hong et al. In this manuscript the author has shown anti-inflammatory properties of Aster koraiensis against dry eye syndrome in vivo and in vitro model and thus can be used as an effective treatment for dry eye syndrome. I enjoyed reading the manuscript. However, I have some suggestions and questions that need to be addressed and should be included to increase the strength of the paper.
- In the introduction section, the hypothesis and aim of the study should be clearly stated.
- We modify the paper to give a reviewer a more explicit description of what the reviewers pointed out. In addition to the Line 81 part, we modified and described the hypothesis, and the aim of this research is simple and straightforward.
- The materials and methods section should be written in more detail for example how much the dried-plant was used for extraction, how much volume of ethanol was used for maceration, what was the volume of A, koraiensisethanol extracts (AKE) was used at the beginning, and how much was at the end after concentrating by evaporation.
- We change the manufacturing method of koraiensis ethanol extract to be more specific.. More specifically, with 95% ethanol for food grade, 250 g of dried A. koraiensis is placed in 2 L of ethanol and stored at room temperature for 3 days. After doing this work in three flasks, use filter paper to remove the residue from each solution. Then, after concentrating using an evaporator, 200 mL of the extract is freeze-dried to produce an extract powder. The final extract powder shows about 75 g and 10% extraction efficiency.
- Within the animal experiment and induction of DE model section under Materials and methods, please mention clearly about all the five groups as it doesn’t say anything about the control group shown in all the figures and tables. Please mention clearly what is this control group as the author only mentions that Groups of mice were orally administered with AKE once per day at concentrations of 0 (as vehicle control), 10, 50, or 100 mg/kg. I assume that this vehicle group is DE. Please correct.
- We modify the paper to be able to more clearly explain methods that can be confusing in the method of animal experiments. The processing conditions for the animal model are as follows. Experimental DE in mice was achieved by twice-daily intraperitoneal (i.p.) injection of 200 μL (2.5 mg/mL) of PBS-diluted scopolamine (Sigma-Aldrich). Groups of mice were orally administered with AKE once per day at concentrations of 0 (as vehicle control), 10, 50, or 100 mg/kg in 200 μL EtOH. In the control group, only PBS buffer was injected without scopolamine. In the case of AKE, 0 mg/kg in200 uL EtOH was administered when orally administered as in the DE. We described it in detail in the revised paper as commented in the review (line 146-152 in the revised manuscript).
- In the material and method section, the author did not mention how the corneal epidermal tissue and lacrimal glands were collected from mice. Please mention it in detail.
- We found that some of the animal tissue extraction methods were missing and modified them to be described in more detail. The animal model extraction process is as follows: Mice are euthanized through cervical dislocation. When removing the cornea, click the mouse's eyelid and use forceps to cut out the eyeball's optic nerve that pops out and removes the cornea. After removing the crystalline lens inside the cornea, store in a freezer at-80o After pulling the mouse's lower jaw with forceps for removal of the lacrimal gland, cut off the epidermis of the pulled part so that the lower part of the mouse's face can be seen. A lacrimal gland located near the lower jaw of the mouse can be secured. The secured lacrimal gland tissue is stored frozen at-80oC. In this regard, we rewrote on line 155-164 in the revised manuscript.
- Please mention how many in vivo and in vitro experiments were performed and whether the in vitroexperiments have been done in duplicate or triplicate in the method section and legend?
- We apologize for not a few details about the number of repeated animal and cell experiments. In the case of the number of animals in the animal experiment, we proceeded to 7 animals and conducted a one-time experiment. In the case of cell experiments, it proceeded through three repeated experiments per experimental group. As the review comments, this part was created by adding it to the figure part.
- In the histology section, it doesn’t mention anything about how the epithelial layer thickness has been measures and quantified (Figure 2A and B), how many areas were selected, single or both eyes were used while quantification, etc. Also, it does not mention how the corneal fluorescence staining was quantified to show corneal damage (shown in Figure 1A and B). Please mention all these things in detail.
- We change the detailed description of histological observation methods that can be confusing as follows. Figure 2A shows a representative photograph of the organization. The thickness of the tissue was measured at 5 locations for each excised tissue. Figure 2B shows the average value of 35 military organizations measured. In the part where Corneal damage is measured, the degree of damage was confirmed after removing both eyes. Fourteen eyeballs were confirmed for each group, and the damage and visible parts were counted one by one, as shown by the arrows in Figure 1A to complete the measurement.
- For the western blot experiment, how many ARPE-19 cells how were used for treatment (TNF/Tg and AKE) in what volume? How long they have been cultured before experiment/treatment? What was used in the control group (mention about vehicle)? At the end of the treatment how the cells were collected for protein extraction? How much protein was used for western blot? Please mention all these in detail.
- We have made the following changes to explain the method created in more detail. 2 x 105 cells per well were cultured in 6 well plates. After culturing for 24 h, 200 uL media solution with 10 μg / mL of TNF-α or 5 μmol / L thapsigargin (Tg) was applied to the cells in the presence or absence of AKE (0, 0.1, 1, and 10 μg / mL) for 12 h. In the Control group case, the experiment was conducted with doubles without any inflammation-inducing factors or AKE. After the treatment was completed, RIPA buffer 80 uL was added for protein extraction, and scrapped cells were secured with scrap par. After centrifuging at 13,000 rpm, the supernatant was taken and quantified (line 203-207 in the revised manuscript).
- How many cells were used for Intracellular calcium release, and VEGF-α secretion experiments and for the QRT-PCR experiment? How long the cells have been before (TNF/Tg and AKE) treatment in what volume? Please mention.
- We apologize for confusion over the methodology of VEGF- α secretion. 1x105 ARPE-19 cells per well were cultured in 96 well plates. In order to measure the VEGF-α secretion, Tg with AKE (0, 0.1, 1, or 10 μg/mL) were dissolved in the medium and treated with 100 μL each, incubated for 24 h, and then doubled and collected (Line 218-222 in the revised manuscript).
- At line 163 it says α-tubulin or GAPDH, while in the whole manuscript only GAPDH has been shown for western blot. Please correct.
- We are very grateful for the meticulous review of paper. We made a mistake during the creation process and did not proceed with the correction of this part. As the reviewer commented, we have removed α-tubulin in the Materials and Methods (Line 183-184 in the revised manuscript).
- In Statistical analysis, please mention in detail which data has been used to analyze by which method.
- We modify the paper so that we can explain the unclearly expressed parts through more certain expressions. Statistical analysis was used on both quantified data to verify significant. In the case of data to which statistics are applied, it is expressed through the legend.
- In Figure 1, please explain the control group (if it is without DE). In legend, it says that AKE was administered at 0 (control; CON), but it seems the DE group has been administered AKE (0, 10, 50, and 100). Please mention clearly about the control group if the mice in the control group were injected PBS by intraperitoneal injection and then given vehicle orally (mention if ethanol). Also please mention the volume used for oral administration.
- We are apologetic that the explanation in Figure Legend led to a lack of understanding of the dissertation. In the legend, "Dry eye (DE) was reportedly induced by scopolamine injection, and AKE was administered at0 (control; CON which was not induced by scopolamine injection, DE), 10 (AKE10), 50 (AKE50), or 100 mg/kg (Fixed a description that can be confusing, such as "AKE100)". And for the control group that did not induce scopolamine, we created materials and methods to administer the PBS buffer when subcutaneous injections were given. It was also created in the description of the dosage and solvent for oral administration of AKE.
- In Figure 1B, please mention the parameter measured/quantified on the y-axis along with the unit of measurement. In Figure 1C, please mention if TBUT is measured in second in legend.
- We apologize for the inadvertent mistake and truly appreciate your meticulous review. Of the content that the review pointed out, first in Figure 1B, we confirmed that the y-axis description was missing, there is a problem in interpreting the data without providing information on the y-axis, and we have corrected it by adding the y-axis in the graph. Next, among the points pointed out, we changed the information about the measurement time in Figure 1C to "(C) Tear breakup time (TBUT) was measured in seconds and analyzed" and corrected it to figure legend.
- In Figure 2A, it would be nice to show a line to show the epithelial layer. In Figure 2B, please write corneal epithelial thickness (um) on the y-axis.
- We are grateful for reviewer's good suggestions. We agreed with the review and reconstructed the y-axis description of the figure 2A.
- At lines 263-266 it says “Lacrimal gland histopathology during AKE treatment of DE mice revealed a decrease in the number of infiltrating immune cells in the tissue, compared with the DE mice without AKE treatment (Figure 2C). Additionally, gaps in the glands were narrowed by the administration of AKE (Figure 2C)”. It does not match with the data shown in Figure 2C (which seems to be mRNA levels of various cytokines in corneal extract). Please correct.
- We change to parts that can confuse the interpretation. As commented in the review, in Figure 2C it is mRNA levels of various cytokines in corneal epithelium. The content of the interpretation of the result is the content of the organizational change of the lacrimal glands, and the appropriate data is figure3. On the other hand, we corrected it in the text.
- Figure 3A and B does not match with the text mentioned under AKE suppressed immune responses in the corneal epithelium and lacrimal glands of DE model mice. Also, mention Figure 3C in the result section. Please correct.
- We change to unnecessary explanations that can confuse the interpretation of the results. The parts that were misinterpreted in the text have been changed as follows. In particular, IFN-γ and MMP-9 were remarkably attenuated, even at a low dose (10 mg/kg) of AKE (Figure 2C, 3A). In lacrimal glands, AKE suppressed the expression of inflammatory cytokines IL-1β, TNF-α, IFN-γ, and TGF-β (Figure 3B). Especially, when 50 mg/kg of AKE was administered, cytokine expression did not decrease in proportion to the concentration in lacrimal gland tissues. In addition, the explanation of the figure was explained as in the advanced 3.3 Histological alterations of the corneal epithelial and lacrimal glands following AKE treatment in the DE mouse model (Line 291-297, 316-323 in the revised manuscript).
- For the QRT-PCR quantification data (Figure 2C, Figure 3A, and Figure 4A) please write if it is a relative expression of the mRNA gene for specific cytokines or mRNA levels as fold change on the y-axis to make it more clearly. Please correct in the graphs and legends.
- We are grateful for reviewer's good suggestions. We agreed with the review and reconstructed the y-axis description of the Figure 2C, 3A, and 4A and legend.
- Please mention for all the western blot quantification results on the y-axis if it is a ratio of the expression level of protein of interest to the housekeeping gene on the graph and in legend.
- We are grateful for reviewer's good suggestions. We agreed with the review and reconstructed the y-axis description of the Figure 2C, 3B, and 4A and legend.
- It would be nice to show the quantification data for the ratio of p-JNK/JNK, p-P38/P38, and p-ERK/ERK for Figure 4B in addition to the western blot image.
- We appreciate the opinions that can improve the quality of our dissertation. Make sure to include it in supplement data in relation to the contents of paper.
- Based on the materials and methods section it seems that Intracellular calcium level was performed in vitro, while in Figure 4D it mentions n = 7 mice per group. Please correct.
- We apologize for creating the wrong figure legend. As the review comments, we have completed the correction work for this part.
- At line 310, it is not clear what is “3.5 AKE inhibited inflammatory response in lacrimal gland”. Please correct.
- We are grateful for reviewer’s meticulous comments on the review. In the case of "3.5 AKE inhibited inflammatory response in lacrimal gland", it is the title of the resulting part. This part was mistakenly entered into Figure Legend during the editing process. Probably due to a system error. Thank you for pointing out the problems that we haven't seen until the end (Line 342 in the revised manuscript).
- At line 38 DE should be written as Dry eye (DE), please correct.
- We are grateful for reviewer’s detailed comments on the parts we have not confirmed. For this part, I created a description of the abbreviation (Line 38 in the revised manuscript).
- At line 84 make koraiensis italics.
- We are grateful for reviewer's detailed comments. However, this part is italicized in the title, so in the case of koraiensis, it is not italicized.
- Please mention the legend below Table 4 (as shown in Table 3).
- We are grateful for pointing out the parts we omitted in the process of creating. This part has been revised to reflect the reviewer's opinion.
- At lines 175-176 it says that “The expression of the target genes was normalized to the expression of GAPDH”, while at lines 278 and 301 it says β-Actin was used as an internal control for QRT-PCR, and at line 355 it says GAPDH was used as an internal control. Please correct and modify Table 2 accordingly.
- We apologize for the confusion in providing information. We have updated the materials and methods and table 2 information in the revised paper.
- In discussion it would be better not to write Figure as () at the end of the sentence at line 383, 385, 393, 394, 408, 411, and 421.
- We appreciate the reviewer’s comment in strengthening the quality of our manuscript and totally agree with the comment. We have corrected all of the issues pointed out.
- The authors should also discuss the limitations of the present study.
- We are very grateful to improve the quality of the paper through the comments in the review. At present, we are not proceeding with application tests to the human body at the limit of our research. We added a limit to this in the conclusion part. Through this research, we have determined that the AKE we have discovered can be used as an actual food material if it is being tested for application to the human body in the future (Line 472-477 in the revised manuscript).
Reviewer 4 Report
The article is interesting and it might be published after few experiments.
Specific comments:
- Does the compound induce toxicity in mice? I would suggest to measure transaminases, hemochrome in mice after different treatment protocol and doses. At least, two strategies (acute and cheonic treatments) should be tested.
- The conclusion are overstated. The results have been performed in mice and cells, not humans. This aspect has to be addressed also in the discussion.
Author Response
The article is interesting and it might be published after few experiments.
- Does the compound induce toxicity in mice? I would suggest to measure transaminases, hemochrome in mice after different treatment protocol and doses. At least, two strategies (acute and cheonic treatments) should be tested.
- We are grateful for the reviews for their sincere advice for improving the paper. Toxicity assessment of AKE was performed indirectly through cell experiments (MTT assay). However, we did not evaluate toxicity tests such as genetic toxicity and repeated administration in animals, which is the essential stage of evaluating its health effects in a human test. We agree with what requires additional experimentation with this, or the current paper states the benefits of AKE and its expectations. In the future, in order to apply it to actual food materials, we plan to conduct application tests and toxicity evaluations on the human body. In this study, the possibility was confirmed, and toxicity evaluation will be published again when used as a food material in the future.
- The conclusion is overstated. The results have been performed in mice and cells, not humans. This aspect has to be addressed also in the discussion.
- We are very grateful for the delicate review of our paper. We agree that our conclusions are so dramatic. To emphasize the possibility of, we did not mention the limits of our research. We have made additional statements about the limits and potential for future research (Line 472-477 in the revised manuscript).
Round 2
Reviewer 3 Report
Manuscript ID: nutrients-958875
Type of manuscript: Article
Title: In vivo anti-inflammation potential of Aster koraiensis extract for dry eye syndrome by the protection of ocular surface
Authors: Sung-Chul Hong, Jung-Heun Ha, Inhae Kang, Sang Hoon Jung, Jin-Chul Kim *
Dear Author,
Thanks for your response and for incorporating the changes in the manuscript. I still find that there are some important changes that need to be addressed in the given manuscript for further improvement.
- The response to the 6th point should be mentioned in thoroughly the manuscript under the histology section by mentioning that “The thickness of the tissue was measured at 5 locations for each excised tissue” and how the quantification was performed to show the Figure 2B by saying that the results were shown by showing an average value of 35 military organizations measured.
- Please mention how the corneal fluorescein staining was performed and quantified in detail under a new heading as corneal damage before histology by adding that “the degree of corneal damage was confirmed after removing both eyes. Fourteen eyeballs were confirmed for each group, and the damage and visible parts were counted one by one.” Please mention how the quantification was performed in Figure 2B showing the corneal fluorescence score and what does the corneal fluorescence score mean? Please explain.
- In lines 291-293 the author mentioned that “Lacrimal gland histopathology during AKE treatment of DE mice revealed a decrease in the number of infiltrating immune cells in the tissue, compared with the DE mice without AKE treatment (Figure 3C). In Figure 3C, it would be nice to show the infiltrating immune cells using an arrow.
- Did the author perform any specific staining to verify what these infiltrating immune cells (in Figure 3C) were in the lacrimal gland? Please comment.
- Please mention how many cells were used for Intracellular calcium release and RT-qPCR experiment and how long the cells have been cultured before treatment and make sure that this information is present for each experiment (Intracellular calcium release, RT-qPCR, VEGF- α and western blot).
- In line 284 under 3.3 section, the author mentioned “Histological alterations of the corneal epithelial and lacrimal glands following AKE treatment in the DE mouse model”. Please show the images of Figure 2A, B, and Figure 3C together as Figure 2A, B, and C respectively and then mentioning those in the results accordingly.
- In line 310 under 3.4 section, the author mentioned “AKE suppressed immune responses in the corneal epithelium and lacrimal glands of DE model mice”. In this part Figure 2C, Figure 3A and, B should be presented as Figure 3A, B, and C respectively, and should be mentioned in the results accordingly.
- In line 318-319 it says “In lacrimal glands, AKE suppressed the expression of inflammatory cytokines IL-1β, TNF-α, IFN-γ, and TGF-β (Figure 3B), while Figure 3A but not Figure 3B is showing the cytokine levels. Please correct.
- For all western blot image please write p-IkB and total IkB, p-NF-kB, and total NF-kB, b-actin, GAPDH, COX1, COX2) in front of the respective blot (one the right or left side) in the upper and lower panel for respective images to make it more clear and easy to understand.
- In line 352 (Figure 3A), please correct as it should be Figure 3B.
- In line 351, please delete the word arose.
- In line 393 and 483 please remove Each bar represents the mean ± SD of n = 3 experiments per group and write that data from three independent experiments have been presented as bar graphs showing mean ± SD.
Author Response
- The response to the 6thpoint should be mentioned in thoroughly the manuscript under the histology section by mentioning that “The thickness of the tissue was measured at 5 locations for each excised tissue” and how the quantification was performed to show the Figure 2B by saying that the results were shown by showing an average value of 35 military organizations measured.
- We appreciate for your great interest in the manuscript. We want to elaborate on the experimental methods performed in connection with tissue observation as follows. The area we measured by looking for three points from the center to both ends and taking two additional points on the left and right between the other middle and both ends. Therefore, we selected and measured 35 points per experimental group.
- Please mention how the corneal fluorescein staining was performed and quantified in detail under a new heading as corneal damage before histology by adding that “the degree of corneal damage was confirmed after removing both eyes. Fourteen eyeballs were confirmed for each group, and the damage and visible parts were counted one by one.” Please mention how the quantification was performed in Figure 2B showing the corneal fluorescence score and what does the corneal fluorescence score mean? Please explain.
- We apologize for the inadequacy of the detailed description of what we have experimented with to derive the data results. The process we performed to compare the degree of fluorescence was as follow. After saving the degree of fluorescence as an image file, we used the image J program to determine where the fluorescence measurement value can be counted. The counted values were compared and analyzed by the difference between each experimental group (line 156 - 158 in the revised manuscript).
- In lines 291-293 the author mentioned that “Lacrimal gland histopathology during AKE treatment of DE mice revealed a decrease in the number of infiltrating immune cells in the tissue, compared with the DE mice without AKE treatment (Figure 3C). In Figure 3C, it would be nice to show the infiltrating immune cells using an arrow.
We appreciate for the interest in our dissertation. The answer is that many immune cells move by the Gap junction (Biochimica et biophysica Acta (BBA)- Biomembranes, Vol 1662, issue 1-2, 102-112, 2004.). According to the reference paper, it can be judged that the influx of immune cells increased in the situation where the Gap junction collapsed, as shown in the data. Those were inferred without additional staining or measurement, we used the expression "Lacrimal gland histopathology during AKE treatment of DE mice revealed a decrease in the number of infiltrating immune cells in the tissue, compared with the DE mice without AKE treatment (Figure 3C)”. We are planning to do the staining for identifying each immune cell infiltrated in the DE tissue and developing the method. After showing the data to the pathologist, we confirmed that those cells in the DE tissue were immune related cells.
- Did the author perform any specific staining to verify what these infiltrating immune cells (in Figure 3C) were in the lacrimal gland? Please comment.
We did not conduct additional staining, but confirmed the data with pathologist as mentioned above.
- Please mention how many cells were used for Intracellular calcium release and RT-qPCR experiment and how long the cells have been cultured before treatment and make sure that this information is present for each experiment (Intracellular calcium release, RT-qPCR, VEGF- α and western blot).
- In in vitro experiments, cells were cultured for 24 hours and adjusted to the appropriate cell number for each. In RT-qPCR and western blot, all cells were collected and tested after each culture and treatment (line 210 - 212 in the revised manuscript).
- In line 284 under 3.3 section, the author mentioned “Histological alterations of the corneal epithelial and lacrimal glands following AKE treatment in the DE mouse model”. Please show the images of Figure 2A, B, and Figure 3C together as Figure 2A, B, and C respectively and then mentioning those in the results accordingly.
- We apologize for the confusion in the interpretation and description of the data. However, to explain the data we presented efficiently, it would be better to separate the data as Figure 2A and B with Figure 3C.
- In line 310 under 3.4 section, the author mentioned “AKE suppressed immune responses in the corneal epithelium and lacrimal glands of DE model mice”. In this part Figure 2C, Figure 3A and, B should be presented as Figure 3A, B, and C respectively, and should be mentioned in the results accordingly.
- We appreciate for your comment to improve the manuscript. However, it would be easier for the reader to understand better way when we put as the way we submitted. We thought it would be better to see Figure 2C and 3A simultaneously, so we made the following expression. Besides, in the section 3.5, we described for Figure 3B. We actually considered as you pointed for the first time, but we separated during manuscript preparation.
- In line 318-319 it says “In lacrimal glands, AKE suppressed the expression of inflammatory cytokines IL-1β, TNF-α, IFN-γ, and TGF-β (Figure 3B), while Figure 3A but not Figure 3B is showing the cytokine levels. Please correct.
- We apologize for any expressions that can cause problems in the interpretation of the data. This part is corrected (line 317 - 319 in the revised manuscript).
- For all western blot image please write p-IkB and total IkB, p-NF-kB, and total NF-kB, b-actin, GAPDH, COX1, COX2) in front of the respective blot (one the right or left side) in the upper and lower panel for respective images to make it more clear and easy to understand.
- We corrected as you pointed (Figure 3B in the revised manuscript).
- In line 352 (Figure 3A), please correct as it should be Figure 3B.
- We fixed the part accordingly (line 350 - 353 in the revised manuscript).
- In line 351, please delete the word arose.
- We deleted the word “arose”
- In line 393 and 483 please remove Each bar represents the mean ± SD of n = 3 experiments per group and write that data from three independent experiments have been presented as bar graphs showing mean ± SD.
- As you recommended, we changed the sentence accordingly.
Reviewer 4 Report
No further comments.
Author Response
We appreciate for reviewing the revised manuscript.